# Introducing the Dutch Quality Registry for Acute Internal Medicine (DRAIM): Method of development and opportunities of use from a single-centre pilot study

Marleen G. A. M. van der Velde[ID][1,2☯]*, Elisabeth M. Mols[1,2☯], Jelmer Alsma[ID][3], Prabath W.B. Nanayakkara[4], Harm R. Haak[1,2☯], Marjolein N. T. Kremers[2,5☯], on behalf of the research consortium acute medicine ORCA[¶]

1 Department of Internal Medicine, Máxima MC, Veldhoven/Eindhoven, the Netherlands, 2 Department of Health Services Research, and CAPHRI School for Public Health and Primary Care, Aging and Long Term Care, Maastricht, the Netherlands, 3 Department of Internal Medicine, Erasmus University Medical Centre, Rotterdam, the Netherlands, 4 Section General Internal Medicine unit Acute Medicine, Amsterdam Public Health Research Institute, Department of Internal Medicine, Amsterdam University Medical Centre, Amsterdam, the Netherlands, 5 Emergency Department, Erasmus University Medical Centre, Rotterdam, the Netherlands

☯ These authors contributed equally to this work.
¶ Complete membership of ORCA can be found in the Acknowledgments.
* marleen.van.der.velde@mmc.nl

## Abstract

### Background

The Dutch Quality Registry for Acute Internal Medicine (DRAIM) aims to evaluate and improve the quality of care for internal medicine patients presenting in the Dutch acute care chain. Using an online dashboard, comprising organizational, process, and patient characteristics, DRAIM provides insight into local institutional characteristics and enables comparisons with a benchmark. The aim is to identify potential bottlenecks, develop of interventions to optimize acute care and to create a learning environment.

### Methods

DRAIM is a multicenter quality registry. However, this study reports findings from a single-centre pilot conducted in one teaching hospital. Data for this cohort was collected from November 2019 to June 2023, including patients presenting in the Emergency Department (ED) for internal medicine. Data was extracted from the Electronic Health Record, including patient and organizational characteristics, such as presenting complaint, and process- and outcome measures, such as length of stay in the ED (LOS-ED).

**Data availability statement:** Data (in Dutch) is available upon reasonable request. Researchers can visit the website (https://www.draim.nl/onderzoek/) for information on submitting a request for data and/or samples.

**Funding:** In 2017, DRAIM was developed as a pilot project with funding from the "Stichting Kwaliteitsgelden Medisch Specialisten" (SKMS). However, funding from SKMS ceased at the end of 2022. Since 2023, DRAIM has been supported with funding provided by Zorgverzekeraars Nederland (ZN). There was no additional external funding received for this study.

**Competing interests:** The authors have declared that no competing interests exist.

## Results

A total of 6,071 patients aged ≥65 years were included. Older adults were more likely to experience adverse outcomes, including longer LOS-ED, increased hospital admissions, extended hospital stays, and higher in-hospital mortality rates. Additionally, 2.366 patients (46.6%) presented with general malaise. Compared to patients with other presenting complaints, those with general malaise had longer LOS-ED and higher hospital admission rates. These findings prompted the development and implementation of a standardized care-pathway for patients with non-specific complaints in the ED.

## Conclusion

This single-centre pilot demonstrates the potential of DRAIM to provide valuable insights into acute-care processes and identify bottlenecks in patient flow. While findings cannot yet be generalised nationally, they highlight important trends, such as the vulnerability of older patients and those presenting with non-specific complaints. Future use of DRAIM for multicentre studies will validate these observations and facilitate the evaluation and improvement of the Dutch acute-care network.

## Introduction

In the Netherlands, a significant proportion of the Emergency Department (ED) workload comprises patients presenting for internal medicine. The ED fulfils a pivotal function in risk stratification and the establishment of initial diagnosis and treatment for these patients. The organization of ED care alongside external factors such as financial resources and demographics, are identified as factors influencing ED performance [1,2]. Therefore, the evaluation of organizational, structural and patient characteristics in the ED can be used to improve quality and enhance patient safety. Nationwide quality registries on trauma or surgical interventions are well-established in the Netherlands, in contrast, evaluation of ED care for internal medicine patients is not structurally implemented [3,4].

The organization of Dutch EDs is heterogeneous. Staffing for example, relies on local organization and working agreements, and can involve residents and specialists of different specialties in various compositions [1]. Given the multidisciplinary staffing of EDs, patients are directly triaged for a specific medical specialty based on their presenting complaints. Patients treated by the specialty of internal medicine, compared to other patient groups, are characterized by a higher age, multimorbidity, polypharmacy and often undifferentiated complaints [5]. As a result, the case complexity is high, requiring additional diagnostics and multidisciplinary care.

Additionally, in contemporary emergency care, an increasing proportion of the population consists of undifferentiated patients, characterized by older age, comorbidity, polypharmacy and non-specific complaints (NSC). These patients often visit the ED due to exacerbations of chronic conditions and are typically treated by an internist.

In 2018, they represented 20% of all patients in Dutch EDs; a percentage that is expected to rise considering the ageing population and increasing prevalence of chronic diseases [6].

The development of the Dutch Registry for Acute and Internal Medicine (DRAIM) offers a valuable resource for evaluating the quality of the acute care pathway. This registry comprises data regarding patient characteristics, such as triage category, severity of disease and comorbidities, combined with process data including length of stay in the ED (LOS-ED) and outcome data such as in-hospital mortality, ED revisits or readmissions. The long-term objective of this quality registry is to engage all EDs across the Netherlands, establishing a robust foundation for enhancing quality of care. Using the DRAIM, we aim to provide insight into the growing and complex patient population using acute care pathways. By comparing various hospitals, organizational structures and specific patient groups, we encourage the enhancement of self-learning capabilities for the purpose of quality improvement. The ultimate goal of DRAIM is to improve the quality of acute care and facilitate delivery of the right care in the right place. In this study, we demonstrate the use of DRAIM for quality evaluation by assessing the association between presenting complaint and emergency department length of stay and hospital admission among older internal medicine patients, adjusting for relevant patient- and presentation-related characteristics.

## Methods

### Study design and setting

We conducted a retrospective single-centre pilot cohort study using data from DRAIM at the Máxima Medical Centre, a teaching hospital in the Netherlands with a 15-bed ED and an Acute Medical Unit. This hospital was the first to fully automate data extraction and submission to the national quality registry.

All consecutive patients aged ≥65 years who presented to the ED with internal medicine-related complaints were included in the quality registry, including subspecialties such as infectious diseases, vascular medicine, geriatric care, endocrinology, nephrology, acute medicine, haematology, oncology, immunology and allergology. Patients under the age of 18 years were excluded.

Patients were stratified into three age categories: 65–74 years, 75–84 years, and ≥85 years. Presenting complaints were categorised as specific or non-specific, with "general malaise" classified as non-specific. While general malaise is not strictly synonymous with the definition of nonspecific complaints found in the literature, we assumed that a considerable proportion of these patients would fit the NSC criteria defined by Nemec et al, due to the absence of a more specific presenting complaint [7].

### Data collection

Clinical and administrative data were extracted quarterly from the electronic health record (EHR) and financial records and uploaded to a secure portal managed by DRAIM. Data quality was ensured by an external data processor (as defined under GDPR) through data sampling, checking the ranges of response options and detection of missing values. The collection of the data as described has been approved by the Medical Ethical Committee of the Máxima MC. A waiver for informed consent was obtained because of the observational nature of the registry.

This cohort includes data collected between November 2019 and June 2023. Data access was granted in August 2023. At no point during or after data collection did we have access to participants. A complete list of recorded variables is provided in S1 Table.

The selection of variables was based on scientific literature, identification of existing quality indicators in acute care, and expert opinions derived via focus groups and online consultation amongst internists and emergency physicians [8–16]. Clinical details such as vital signs, triage category and presenting complaint were abstracted directly from the EHR. Missing physiological values were recorded as such and assumed to be in the normal range for analysing purposes.

Results of laboratory tests were extracted from the EHR, missing values were recorded as such. Diagnosis and co-morbidities were derived from the financial data records or directly transferred from the EHR.

### Computed variables

Disease severity was assessed using the Modified Early Warning Score (MEWS) and the AVPU scale (Alert, Verbal, Pain, Unconscious) [17]. The age-adjusted Charleson Comorbidity Index (CCI) was calculated based on age and the international Classification of Diseases, 10th edition (ICD-10) codes in the EHR [18–20].

### Statistical method

Baseline characteristics were summarized using descriptive statistics. Continuous variables are presented as means with standard deviations when normally distributed, and as medians with interquartile ranges otherwise. Categorical variables are presented as counts and percentages. Analyses were performed using complete cases, as the proportion of missing data was limited. Group comparisons employed chi-square or Fisher's exact tests for nominal categorical variables, and Mann–Whitney U or Kruskal-Wallis tests for ordinal variables. Continuous variables were compared using ANOVA.

The association between presenting complaint and emergency department length of stay (LOS-ED) was assessed using multivariable linear regression, with log-transformed LOS-ED due to right-skewness. Results are reported as regression coefficients (B) with 95% confidence intervals (CI). Hospital admission was analyzed using multivariable logistic regression, with results reported as odds ratios (ORs) and 95% confidence intervals. Both models were adjusted a priori for age, sex, urgency category (Netherlands Triage System (NTS), U0–U5, with U0 indicating the highest and U5 the lowest urgency)comorbidity burden as measured by Charlson Comorbidity Index (CCI), mode of arrival and number of medications used prior to the ED visit. NTS and comorbidity burden were entered as categorical variables with U3 (moderate urgency) and no comorbidity as reference categories; mode of arrival was dichotomized as ambulance versus own transport. Linearity of age was assessed by including a quadratic age term, and effect modification by presenting complaint was evaluated using an interaction term between age and general malaise. As neither term was statistically significant, they were not retained in the final models. Multicollinearity was assessed using variance inflation factors. Model fit for the linear regression was assessed using adjusted $R^2$. For the logistic regression model, model fit was assessed using −2 log likelihood, Nagelkerke $R^2$, and the Hosmer–Lemeshow test. Statistical significance was defined as a two-sided p-value <0.05. Analyses were conducted using IBM SPSS Statistics (version 29).

### Ethics and patient involvement

This study has been approved by the Medical Ethical Committee of Máxima MC, with a waiver for informed consent because of the observational nature. A representative from the Dutch Patient Federation contributed to the development of DRAIM to ensure patient representation.

## Results

The cohort consisted of 11.118 patients registered for internal medicine between November 2019 and June 2023 (Fig 1).

Patient characteristics are presented in Table 1. Among the cohort, 6.071 persons were aged ≥ 65 years (54.6%). The average age off all patients was 78.3 ± 7.7 years, sex was equally distributed. Of these patients, 5.272 patients had a CCI-score ≥5 (86.2%), while only 28.4% used ≥5 medications.

### Baseline characteristics and outcome measurements: Age-categories

Differences in patient characteristics was observed when comparing age categories, e.g., 65–74 years, 75–84 years and ≥85 years. As age increased, a higher proportion of patients were female, had higher CCI-scores and presented more often with general malaise as presenting complaint and were triaged into lower urgency-categories.

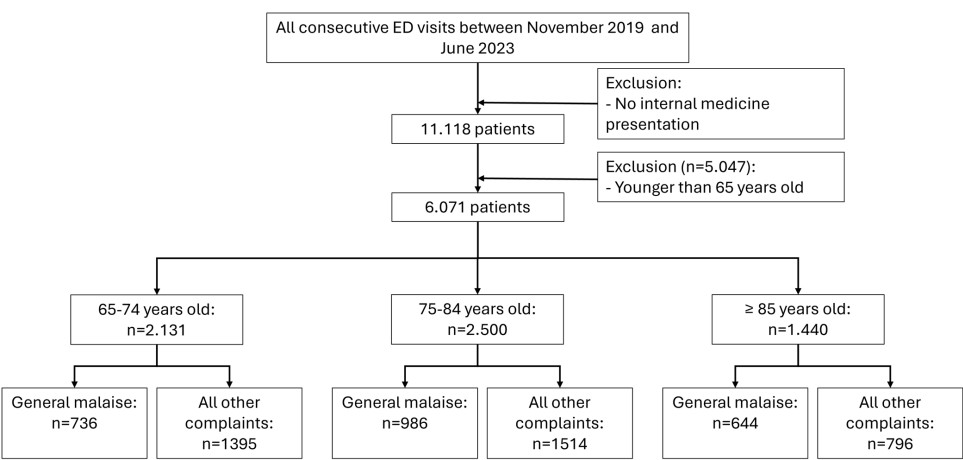

**Fig 1. Patient Selection and Cohort Flow.**

In addition, the outcome measurements are presented in Table 2. The median length of stay in the ED (LOS-ED) was 181 minutes (Q1-Q3; 141–231). In 3.711 patients (61.1%) the ED-visit resulted in hospital admission, with a median length of admission of 6 days (Q1-Q3: 3–10). During hospitalisation 365 patients died (6.0%).

Missing data was marginal, specifically; CCI n = 799 (11.9%), way of referral n = 445 (6.6%), presenting complaint n = 265 (4.0%), hospital admission n = 198 (3.0%), revisit within 7 days n = 198 (3.3%).

## Baseline characteristics and outcome measurements: Presenting complaints

Among patients aged ≥65 years, the most common presenting complaints were general malaise, shortness of breath, fever, abdominal pain, and lower extremity complaints (S2 Table). General malaise was the most frequent, affecting 35.3% of patients and increasing with age from 36.1% (65–74 years) to 46.6% (≥85 years).

In total, 2.366 patients (46.6%) presented with general malaise. Compared with other complaints, these patients were older (79.0 ± 7.7 vs 77.9 ± 7.7 years, p < 0.001), more often female (53.5% vs 48.0%, p < 0.001), and had higher comorbidity (CCI ≥ 5: 74.3% vs 66.0%, p < 0.001) (Table 3). They more frequently arrived by own transport (58.1%), were referred by their general practitioner (54.7%), visited the ED on weekdays (75.1%), and were most commonly triaged as U2 or U3 (41.7% and 34.8%, respectively).

Additionally, patients presenting with general malaise experience more adverse outcomes, such as a longer LOS-ED (184 vs 178 minutes, p < 0.001), a higher hospital admission rate (68.2% vs 60.0%), p < 0.001) and fewer ICU admissions (1.3% vs 3.0%, p < 0.001) (Table 4). No differences in revisits (12.4% vs 12.6%, p = 0.894) or in-hospital mortality (6.3% vs 5.9%, p = 0.524) were found.

## Association of presenting complaints with ED outcomes

In multivariable analyses (Table 5), general malaise was independently associated with both prolonged emergency department length of stay (LOS-ED) and hospital admission. After adjustment for age, sex, urgency category, comorbidity burden, mode of arrival and medication use, patients presenting with general malaise had a longer LOS-ED (B = 0.046, 95% CI 0.022–0.070), corresponding to an approximately 4.7% increase in LOS-ED.

In addition, general malaise was associated with a 51% higher odds of hospital admission (OR 1.51, 95% CI 1.34–1.71). Increasing age, higher urgency categories, and arrival by ambulance were also independently associated with both outcomes.

**Table 1. Baseline characteristics of included patients.**

| | All patients (≥65 years old) (n=6071) | 65-74 years old (n=2131) | 75-84 years old (n=2500) | ≥ 85 years old (n=1440) | p-value |
|---|---|---|---|---|---|
| **Age** | 78 (72–84) | 70 (68–73) | 79 (77–82) | 88 (86–91) | <0.001 |
| **Sex** | | | | | |
| Male | 3028 (49.9%) | 1188 (55.7%) | 1219 (48.8%) | 621 (43.1%) | <0.001 |
| Female | 3043 (50.1%) | 943 (44.3%) | 1281 (51.2%) | 819 (56.9%) | |
| **Amount of medication** | | | | | |
| 0 | 1452 (23.9%) | 524 (24.6%) | 591 (23.6%) | 337 (23.4%) | 0.129 |
| 1-2 | 1927 (31.7%) | 695 (32.6%) | 776 (31.0%) | 456 (31.7%) | |
| 3-4 | 967 (15.9%) | 322 (15.1%) | 416 (16.6%) | 229 (15.9%) | |
| ≥5 | 1725 (28.4%) | 590 (27.7%) | 717 (28.7%) | 418 (29.0%) | |
| **CCI, categorical** | | | | | |
| None (0) | 1 (0%) | 0 (0%) | 1 (0.0%) | 0 (0%) | <0.001 |
| Mild (1–2) | 174 (2.9%) | 171 (9.1%) | 2 (0.1%) | 1 (0.1%) | |
| Moderate (3–4) | 1450 (23.9% | 598 (31.7%) | 499 (23.3%) | 353 (28.4%) | |
| Severe (≥ 5) | 5272 (86.2%) | 1116 (59.2%) | 1641 (76.6%) | 890 (71.5%) | |
| *Missing 11.9%* | | | | | |
| **Way of referral** | | | | | |
| Own transport | 3292 (54.6%) | 1394 (65.8%) | 1322 (53.3%) | 576 (40.3%) | <0.001 |
| Ambulance | 2737 (45.4%) | 725 (34.2%) | 1159 (46.7%) | 853 (59.7%) | |
| **Referrer** | | | | | |
| GP | 2977 (49.0%) | 1009 (52.1%) | 1225 (52.7%) | 743 (54.4%) | <0.001 |
| Self-referral | 2256 (37.2%) | 747 (38.5%) | 939 (40.4%) | 570 (41.8%) | |
| Ambulance | 393 (6.5%) | 182 (9.4%) | 159 (6.8%) | 52 (3.8%) | |
| *Missing 6.6%* | | | | | |
| **Urgency code** | | | | | |
| U0 | 39 (0.6%) | 17 (0.8%) | 17 (0.7%) | 5 (0.3%) | <0.001 |
| U1 | 783 (12.9%) | 285 (13.5%) | 319 (12.8%) | 179 (12.4%) | |
| U2 | 2548 (42.0%) | 948 (44.9%) | 1043 (41.9%) | 557 (38.7%) | |
| U3 | 2399 (39.5%) | 787 (37.3%) | 979 (39.3%) | 633 (44.0%) | |
| U4 | 100 (1.6%) | 27 (1.3%) | 55 (2.2%) | 18 (1.3%) | |
| U5 | 171 (2.8%) | 48 (2.3%) | 77 (3.1%) | 46 (3.2%) | |
| **Presentation on Monday-Friday** | 4701 (77.4%) | 1616 (75.8%) | 1960 (78.4%) | 1125 (78.1%) | 0.088 |

CCI: age adjusted Charlson Comorbidity Index. GP: general practitioner. NA: not applicable. Self-referral encompasses patients who are directed to the ED either by specialists within the hospital or by their own initiative. Data is reported as number of patients (%), mean ± standard deviation or median (Q1-Q3). Percentages are calculated based on the total number of patients in the specific column.

The linear regression model explained 3.4% of the variance in LOS-ED, while the logistic regression model explained 21.3% of the variance in hospital admission. No evidence of multicollinearity was observed. Although the Hosmer–Lemeshow test indicated imperfect calibration for the admission model, this is likely attributable to the large sample size.

## Discussion

In this cohort study we demonstrate that DRAIM provides valuable insights, identifying bottlenecks in the patient journey throughout the acute care pathway, facilitating the evaluation and improvement of acute care quality. We showed

**Table 2. Outcome measurements of included patients.**

| | All patients (≥65 years old) (n = 6071) | 65-74 years old (n = 2131) | 75-84 years old (n = 2500) | ≥ 85 years old (n = 1440) | p-value |
|---|---|---|---|---|---|
| LOS ED, minutes | 181 (141–231) | 176 (129–223) | 180.5 (136–225) | 191 (146–236) | <0.001 |
| Hospital admission, yes *Missing 3.0%* | 3711 (61.1) | 1190 (55.8) | 1532 (61.3) | 989 (68.7) | <0.001 |
| Length of hospital stay, days | 144 (72–240) | 120 (48–192) | 144 (48–240) | 144 (48–240) | <0.001 |
| In hospital mortality, yes | 860 (14.2) | 303 (14.2) | 338 (13.5) | 219 (15.2) | 0.341 |
| ICU admission within 24h, yes | 85 (1.4) | 47 (2.2) | 28 (1.1) | 10 (0.7) | <0.001 |
| Revisit within 7 days not admitted patients, yes *Missing 3.0%* | 163 (7.5) | 79 (9.3) | 51 (5.7) | 22 (5.2) | 0.013 |
| Revisit within 30 days, admitted patients, yes *Missing 3.0%* | 464 (12.5) | 178 (15.0) | 200 (13.1) | 86 (8.7) | <0.001 |

*LOS ED: length of stay Emergency Department. ICU: Intensive Care Unit. Data is reported as number of patients (%), mean ± standard deviation or median (Q1-Q3). Percentages are calculated based on the total number of patients in the specific column..Outcomes also differed by age, with older patients experiencing longer LOS-ED and more frequent hospital admission. Because presenting complaint varied across age groups and may influence diagnostic complexity and care processes, we subsequently examined the association between presenting complaint and LOS-ED.*

that approximately half of the patients presenting for internal medicine in the ED were aged ≥65 years, and one-third presented with general malaise. Patients with general malaise were more often older, female, had more comorbidities, and were triaged into lower urgency categories. Importantly, multivariable analyses showed that general malaise was independently associated with both prolonged emergency department length of stay and higher likelihood of hospital admission, even after adjusting for age, sex, urgency, comorbidity, mode of arrival, and medication use. Other independent predictors included increasing age, higher urgency categories, and arrival by ambulance. These findings emphasize that general malaise identifies a subgroup of patients with increased complexity and vulnerability, contributing to higher resource use in acute care settings. While findings cannot yet be generalized nationally, they highlight important trends, particularly the vulnerability of older patients and those presenting with non-specific complaints.The growing proportion of older patients presents a challenge for clinicians, as they are more likely to present with poorly defined symptoms. General malaise is one such symptom and can be considered a form of nonspecific complaint (NSC). According to Nemec et al., NSC include any complaints that do not fit into a defined category of specific symptoms or signs [7]. While general malaise is not synonymous with NSC as strictly defined in the literature, we assume that a considerable number of these patients fall into this category due to the absence of a more specific complaint. To further support this proposition, the identified characteristics of general malaise patients in this cohort, such as a higher prevalence among females, increased CCI's, more frequent arrivals with their own transport, and frequent referrals by their general practitioner (GP), are comparable to those found in literature of patients presenting with NSC [21,22]. Moreover, they experienced more adverse outcomes, such as a longer LOS-ED and higher hospital admission rates [23–26].

The added value of DRAIM is the ability to provide insights into the patient journey throughout the acute care pathway, facilitating the evaluation and improvement of the quality of acute care. For example, the observed adverse outcomes in older patients presenting with general malaise prompted questions about how to improve care for these patients. In response, a standardized care-pathway was developed and implemented as a pilot in the ED for older patients with NSC [27]. This care-pathway includes standardized history taking and a set of standardized diagnostic measurements, to structure and streamline care for these patients.

**Table 3. Patient characteristics divided in general malaise vs all other presenting complaints.**

| | General malaise (n = 2366) | All other presenting complaints (n = 3705) | p-value |
|---|---|---|---|
| Age | 79.0 ± 7.7 | 77.9 ± 7.7 | <0.001 |
| Sex | | | |
| Male | 1101 (46.5%) | 1927 (52.0%) | <0.001 |
| Female | 1265 (53.5%) | 1778 (48.0%) | |
| Amount of medication | | | |
| 0 | 590 (24.9%) | 826 (23.3%) | 0.006 |
| 1-2 | 793 (33.5%) | 1134 (30.6%) | |
| 3-4 | 354 (15.0%) | 613 (16.5%) | |
| ≥5 | 629 (26.6%) | 1096 (29.6%) | |
| CCI, categorical | | | |
| None (0) | 1 (0.0%) | 0 (0.0%) | <0.001 |
| Mild (1–2) | 50 (2.5%) | 124 (3.8%) | 0.390 |
| Moderate (3–4) | 473 (23.2%) | 977 (30.2%) | 0.005 |
| Severe (≥ 5) | 1511 (74.3%) | 2136 (66.0%) | <0.001 |
| *Missing 11.9%* | | | <0.001 |
| Emergency visit measurements | | | |
| Way of referral | | | |
| Own transport | 1368 (58.1%) | 1924 (52.3%) | <0.001 |
| Ambulance | 985 (41.9%) | 1752 (47.7%) | |
| Referrer | | | |
| GP | 1191 (54.7%) | 1786 (51.8%) | <0.001 |
| Self-referral | 793 (36.4%) | 1463 (42.4%) | 0.105 |
| Ambulance | 194 (8.9%) | 199 (5.8%) | <0.001 |
| *Missing 6.6%* | | | <0.001 |
| Urgency code | | | |
| U0 | 2 (0.1%) | 37 (1.0%) | <0.001 |
| U1 | 122 (5.2%) | 661 (18.0%) | |
| U2 | 1017 (43.0%) | 1531 (41.7%) | |
| U3 | 1120 (47.3%) | 1279 (34.8%) | |
| U4 | 25 (1.1%) | 75 (2.0%) | |
| U5 | 80 (3.4%) | 91 (2.5%) | |
| Presentation on Monday-Friday | 1920 (81.1%) | 2781 (75.1%) | <0.001 |

CCI: Charlson Comorbidity Index. GP: general practitioner. NA: not applicable. Self-referral encompasses patients who are directed to the ED either by specialists within the hospital or by their own initiative. Data is reported as number of patients (%), mean ± standard deviation or median (Q1-Q3). Percentages are calculated based on the total number of patients in the specific column.

## Future recommendations

Our results, along with recent analyses, reveal that ageing patients are characterized by multimorbidity, polypharmacy, and often present with undifferentiated complaints [5,28]. As the population of older patients continues to grow, the prevalence of chronic diseases and comorbidities are expected to rise [29]. Therefore, facilitating the right care in the right place becomes increasingly important for delivering acute care in a sustainable and futureproof manner. It has been suggested that an increasing number of ED visits stem from exacerbations of chronic diseases [6]. Hence, restructuring the acute care chain for chronically ill patients seeking acute care has been proposed to alleviate some of the strain on the ED, such

**Table 4. Outcome measurements divided in general malaise and all other complaints.**

| | General malaise (n = 2366) | All other presenting complaints (n = 3705) | p-value |
|---|---|---|---|
| LOS ED, minutes | 184 (85) | 178 (92) | <0.001 |
| Hospital admission, yes *Missing 3.0%* | 1557 (68.2%) | 2154 (60.0%) | <0.001 |
| Length of hospital stay, days | 6 (8) | 6 (7) | 0.215 |
| In hospital mortality, yes | 217 (5.9%) | 148 (6.3%) | 0.524 |
| ICU admission within 24h, yes | 21 (1.3%) | 64 (3.0%) | <0.001 |
| Revisit within 7 days not admitted patients, yes *Missing 3.0%* | 56 (7.7%) | 107 (7.5%) | 0.838 |
| Revisit within 30 days, admitted patients, yes *Missing 3.0%* | 196 (12.6%) | 268 (12.4%) | 0.894 |

LOS ED: length of stay Emergency Department. ICU: Intensive Care Unit. Data is reported as number of patients (%), mean ± standard deviation or median (interquartile range). Percentages are calculated based on the total number of patients in the specific column.

**Table 5. Multivariable Associations of Patient and Presentation Characteristics with Emergency Department Length of Stay (LOS-ED) and Hospital Admission.**

| Predictor | LOS-ED | | Hospital admission | |
|---|---|---|---|---|
| | B (95% CI) | p-value | OR (95% CI) | p-value |
| General malaise, yes | 0.046 (0.022–0.070) | <0.001 | 1.51 (1.34–1.71) | <0.001 |
| Age (per year) | 0.004 (0.002–0.006) | <0.001 | 1.018 (1.010–1.026) | <0.001 |
| Female sex | 0.008 (−0.015–0.030) | 0.517 | 0.90 (0.80–1.01) | 0.068 |
| Urgency category | | | | |
| U3 | Ref. | | Ref. | |
| U0 | −0.339 (−0.482–−0.196) | <0.001 | 0.71 (0.33–1.51) | 0.373 |
| U1 | 0.028 (−0.010–0.066) | 0.154 | 2.21 (1.79–2.71) | <0.001 |
| U2 | 0.038 (0.013–0.064) | 0.003 | 1.41 (1.24–1.60) | <0.001 |
| U4 | −0.062 (−0.153–0.028) | 0.179 | 0.76 (0.49–1.19) | 0.236 |
| U5 | −0.205 (−0.275–−0.135) | <0.001 | 0.50 (0.35–0.71) | <0.001 |
| Comorbidity (CCI) | | | | |
| None | Ref. | | Ref. | |
| Mild | 0.009 (−0.067–0.086) | 0.813 | 0.39 (0.26–0.57) | <0.001 |
| Moderate | 0.065 (0.025–0.104) | 0.001 | 0.53 (0.42–0.66) | <0.001 |
| Severe | 0.089 (0.054–0.123) | <0.001 | 0.89 (0.72–1.08) | 0.238 |
| Arrival by ambulance | 0.070 (0.046–0.094) | <0.001 | 1.73 (1.53–1.97) | <0.001 |
| Number of medications | −0.003 (−0.006–0.000) | 0.022 | 0.84 (0.83–0.85) | <0.001 |

LOS ED: length of stay Emergency Department.

as the establishment of rapid access clinics. This approach supports the principle of delivering the right care in the right place, although the impacts on the patient journey still need to be evaluated. To evaluate and effectively support these changes, we recommend nationwide participation to strengthen the utilization of DRAIM. This would not only facilitate comparisons among hospitals that have implemented various care pathways for older patients, but also enable ongoing improvements and evaluations to enhance the quality of care.

A profound discussion on defining quality for patients requiring acute internal medicine care, among healthcare professionals, managers, and patients, can help interpret and utilize outcomes to optimize acute care. We recommend to integrate the perceived quality of care by patients in order to combine clinical and organizational outcomes with the patient's perspective, aiming to optimize quality while taking into account different quality domains [30].

### Strengths and limitations

The benefits and usage of a quality registry for patients presenting for internal medicine in the ED are comprehensive. DRAIM can provide information on relevant patient characteristics, including diagnoses (based on ICD-10), disease severity and outcomes. Additionally, it enables the assessment of ED performance by collecting data on the organizational, structural and process characteristics of the ED, and allows for adjustments based on patient characteristics if necessary. Furthermore, changes over time can be objectified following implementation of quality-improvement initiatives. Lastly, comparing outcomes across various EDs may contribute to the identification of bottlenecks in local organizations and provide opportunities for learning and improvement.

Limitations of our study include the following: firstly, a minority of variables in this cohort were incomplete. However, we try to minimize incomplete variables, by providing the local data collection team with feedback reports on missing data. Secondly, data extraction relies on the data registry in the electronic health records (EHR), where diagnoses are primarily based on financial records. This approach can lead to inaccuracies, as seen in cases where patients presenting with acute exacerbations of chronic conditions may be classified under their underlying chronic illness rather than the acute episode itself. Similarly, in the center used for this cohort, patients referred to the ED by their primary medical specialist were registered as self-referral, making it difficult to distinguish them from true self-referred patients. Furthermore, the registry does not capture information on specific treatments received in the ED. Recording all treatments in a standard quality registry is not feasible due to the vast number and complexity of treatment options, as well as the lack of standardized documentation in EHRs. However, as DRAIM continues to develop, we will assess whether treatment details and adherence to guidelines can be incorporated into the registry. Lastly, data is gathered and analyzed with a delay of at least one month. Providing real-time insights would significantly enhance the value of the reported information, allowing for immediate adjustments in the organization of care.

## Conclusion

DRAIM provides valuable insights and identifying bottlenecks in the patient journey throughout the acute care pathway, facilitating the evaluation and improvement of acute care quality. We have demonstrated that older patients presenting to the ED with poorly defined symptoms experience more adverse outcomes, particularly longer LOS-ED and higher hospital admission rates. To further improve acute care, we recommend nationwide implementation of DRAIM to enable benchmarking across hospitals, support evaluation of new care pathways, and integrate patient-reported outcomes, that improvements are guided by both clinical efficiency and the patient perspective.

### Patient and public involvement

A representative from the Dutch Patient Federation is involved in the development and steering committee of DRAIM to ensure patient representation.

## Supporting information

**S1 Table. Dataset of the quality registry acute internal medicine.**
(DOCX)

**S2 Table. Top 5 presenting complaints in the ED.**
(DOCX)

 

## Acknowledgments

We extend our gratitude to Gertjan Mantjes and Samara Jaber from the Netherlands Association of Internal Medicine for their dedicated efforts in organizing this project. Our sincere thanks go to Peter Verhaegh, an information technician at the Máxima Medical Centre, for his valuable contribution in extracting data from the EHR.

Moreover, we express our appreciation to Marco Moesker for his dedication to developing this quality registry and online dashboard from the TTP perspective, as well as to Liora Rodilla and Léonie Schuurmans for their support in project management from the same perspective.

We also wish to acknowledge the contributions of all DRAIM ambassadors from the participating hospitals for their support in the local implementation of DRAIM, including Tanka Minderhoud (University Medical Center Utrecht), Judith Hillen (Isala Hospital Zwolle), and Marijke van Gerwen (Frisius Medical Center Leeuwarden and Femke Jonkers as the ambassador of the Máxima Medical Centre. Additionally, we extend our gratitude to the DRAIM steering committee members: S. Jaber, L. Rodill, L. Schuurmans, D. Sprengers, D. Tjwa.

## Collaboration

DRAIM invites all Dutch hospitals to participate in the quality registry. Participation and the associated connection to the central database, maintenance contracts, central analysis, and use of the dashboard are provided at no cost due to funding.

## Author contributions

**Conceptualization:** Harm R. Haak, Marjolein N.T. Kremers.

**Data curation:** Marleen G.A.M. van der Velde, Elisabeth M. Mols, Marjolein N.T. Kremers.

**Formal analysis:** Marleen G.A.M. van der Velde, Elisabeth M. Mols, Harm R. Haak.

**Methodology:** Marleen G.A.M. van der Velde, Elisabeth M. Mols, Marjolein N.T. Kremers.

**Project administration:** Elisabeth M. Mols.

**Supervision:** Harm R. Haak, Marjolein N.T. Kremers.

**Writing – original draft:** Marleen G.A.M. van der Velde, Elisabeth M. Mols.

**Writing – review & editing:** Marleen G.A.M. van der Velde, Elisabeth M. Mols, Jelmer Alsma, Prabath WB Nanayakkara, Harm R. Haak, Marjolein N.T. Kremers.

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
