## [Editor Report · Decision Letter 0]

4 Mar 2025

PONE-D-25-05039Introducing the Dutch Quality Registry for Acute Internal Medicine (DRAIM): Method of development and opportunities of use.PLOS ONE

Dear Dr. van der Velde,

Thank you for submitting your manuscript to PLOS ONE. After careful consideration, we feel that it has merit but does not fully meet PLOS ONE’s publication criteria as it currently stands. Therefore, we invite you to submit a revised version of the manuscript that addresses the points raised during the review process.

**Thank you authors for the submission.**

This study is very interesting.

There is novelty in this study.

I give some comments and suggestions.

- The method should be clear to replicate the findings.

- The registry should include exact diagnosis (based on ICD), co-morbidity, treatment, and outcome.

- Is it to broad to include all diagnosis in internal medicine ? The board criteria will not focus in the specific treatment and outcome.

- Please begin the discussion with the main findings of your study.

- Please elaborate more the limitation of the study.

- The conclusion should be concise and clear, and only answer the research question.

Please submit your revised manuscript by Apr 18 2025 11:59PM. f you will need more time than this to complete your revisions, please reply to this message or contact the journal office at plosone@plos.org. Please include the following items when submitting your revised manuscript:

We look forward to receiving your revised manuscript.

Kind regards,

Rizaldy Taslim Pinzon

Academic Editor

PLOS ONE

**Journal Requirements:**

1. When submitting your revision, we need you to address these additional requirements. Please ensure that your manuscript meets PLOS ONE's style requirements, including those for file naming. The PLOS ONE style templates can be found at https://journals.plos.org/plosone/s/file?id=wjVg/PLOSOne_formatting_sample_main_body.pdf and https://journals.plos.org/plosone/s/file?id=ba62/PLOSOne_formatting_sample_title_authors_affiliations.pdf 2. Thank you for stating in your Funding Statement:  In 2017, DRAIM was developed as a pilot project with funding from the "Stichting Kwaliteitsgelden Medisch Specialisten" (SKMS). However, funding from SKMS ceased at the end of 2022. Since 2023, DRAIM has been supported with funding provided by Zorgverzekeraars Nederland (ZN). Please provide an amended statement that declares *all* the funding or sources of support (whether external or internal to your organization) received during this study, as detailed online in our guide for authors at http://journals.plos.org/plosone/s/submit-now. Please also include the statement “There was no additional external funding received for this study.” in your updated Funding Statement. Please include your amended Funding Statement within your cover letter. We will change the online submission form on your behalf. 3. In the online submission form, you indicated that your data is available only on request from a third party. Please note that your Data Availability Statement is currently missing the name of the third party contact or institution. Please update your statement with the missing information. 4. One of the noted authors is a group or consortium “research consortium acute medicine ORCA”. In addition to naming the author group, please list the individual authors and affiliations within this group in the acknowledgments section of your manuscript. Please also indicate clearly a lead author for this group along with a contact email address. 5. Your ethics statement should only appear in the Methods section of your manuscript. If your ethics statement is written in any section besides the Methods, please move it to the Methods section and delete it from any other section. Please ensure that your ethics statement is included in your manuscript, as the ethics statement entered into the online submission form will not be published alongside your manuscript.

**Additional Editor Comments:**

Thank you authors for the submission.

This study is very interesting.

There is novelty in this study.

I give some comments and suggestions.

- The method should be clear to replicate the findings.

- The registry should include exact diagnosis (based on ICD), co-morbidity, treatment, and outcome.

- Is it to broad to include all diagnosis in internal medicine ? The board criteria will not focus in the specific treatment and outcome.

- Please begin the discussion with the main findings of your study.

- Please elaborate more the limitation of the study.

- The conclusion should be concise and clear, and only answer the research question.

---

## [Author Response · Author response to Decision Letter 1]

10 Apr 2025

Dear Dr. Pinzon,

We thank you for your effort to assess our manuscript. The feedback on the manuscript was helpful and we feel that this contributed to improve the quality of our paper.

In response to the comments, we have revised the manuscript accordingly, with all changes highlighted using track changes. We have also provided a detailed, point-by-point reply addressing each concern.

Key revisions include a clearer presentation of the study’s aim and methods to enhance replicability. As requested, we have restructured the discussion to first elaborate on the main findings. Additionally, we have expanded the limitations section.

We have also made the necessary adjustments to meet the journal's requirements, including updating our funding statement: ‘In 2017, DRAIM was developed as a pilot project with funding from the "Stichting Kwaliteitsgelden Medisch Specialisten" (SKMS). However, funding from SKMS ceased at the end of 2022. Since 2023, DRAIM has been supported by Zorgverzekeraars Nederland (ZN). No additional external funding was received for this study.’

We believe that the revised manuscript is now suitable for publication in PLOS ONE.

On behalf of all authors,

M.G.A.M. van der Velde & E.M. Mols

Point to point response

Reviewer#1

Thank you authors for the submission.

This study is very interesting.

There is novelty in this study.

I give some comments and suggestions.

We thank you for your effort to assess our manuscript. The feedback on the manuscript was helpful and we feel that this helped us to improve the quality of our paper

Comment 1.1: The method should be clear to replicate the findings.

Reply 1.1: Thank you for your comment. We agree and have revised the manuscript to improve clarity and ensure replicability of the findings. The most significant change is that we reorganized the text to follow a more standard structure: Introduction, Methods, Results, Discussion, and Conclusion. In particular, we have rewritten the Methods section for better clarity, added the statistical methods, and ensured that all methodological content is now included in this section, rather than in the previous "Findings to Date" section. These revisions should facilitate replication of the findings. Additionally, we have rewritten the Results section to focus solely on the results, moving any introductory content to the Introduction section.

See page 4, lines 76-81 for the adjustment of the introduction: ‘Additionally, in contemporary emergency care, an increasing proportion of the population consists of undifferentiated patients, characterized by older age, comorbidity, polypharmacy and non-specific complaints. These patients often visit the ED due to exacerbations of chronic conditions and are typically treated by an internist. In 2018, they represented 20% of all patients in Dutch EDs; a percentage that is expected to rise considering the ageing population and increasing prevalence of chronic diseases (6).

See page 5, lines 99-117, for adjustments to the method section: ‘The Máxima Medical Centre was the first participating hospital to have fully automated its data-extraction and submission. Therefore, we used the data of this hospital as a showcase with a retrospective single-center cohort study design. Máxima MC – Veldhoven is a teaching hospital in the Netherlands. The ED of the participating hospital contains 15 treatment bays, including one trauma-room and one shock-room. An Acute Medical Unit is present. Eligible patients for inclusion in the quality registry were all patients who presented or were admitted to the specialty of internal medicine from the ED, including its subspecialties such as infectious diseases, vascular medicine, geriatric care, endocrinology, nephrology, acute medicine, hematology, oncology, immunology and allergology. Patients under the age of 18 years were excluded.

For this study, all consecutive patients aged ≥65 years who presented to the ED with internal medicine-related complaints were included. Patients were stratified into three age categories: 65–74 years, 75–84 years, and ≥85 years. To distinguish between patients with specific versus nonspecific presentations, we categorized patients with the primary presenting complaint general malaise as non-specific and all other primary presenting complaints as specific. . While general malaise is not strictly synonymous with the definition of nonspecific complaints found in the literature, we assumed that a considerable proportion of these patients would fit the NSC criteria defined by Nemec et al, due to the absence of a more specific presenting complaint (1).’

See page 8-9, lines 151-168: Statistical method. Baseline characteristics of the participants were analyzed using descriptive statistics. Continuous data were presented as mean and standard deviation (SD) or median and interquartile range (IQR), and nominal data were presented as total amount and percentage. To compare the baseline characteristics between groups, p-values were calculated by using Pearson’s chi-square or Fisher’s exact test for categorical variables, and the independent-samples t-test or Mann-Whitney U test for continuous variables. A significance level of 5% was used for all statistical tests. Statistical analyses were performed using the software Statistical Package for the Social Sciences (SPSS), version 27 (IBM Corp, New York, USA). Patient and public involvement. A representative from the Dutch Patient Federation is involved in the development of DRAIM to ensure patient representation. Ethics. The collection of the data as described above has been approved by the Medical Ethical Committee of the Máxima MC. A waiver for informed consent was obtained because of the observational nature of the project.’

Comment 1.2: The registry should include exact diagnosis (based on ICD), co-morbidity, treatment, and outcome.

Reply 1.2: Thank you for your feedback. As stated in the article, our data is sourced from the Dutch Registry for Acute and Internal Medicine (DRAIM). The methods section provides a comprehensive list of the recorded variables (Table 1). Diagnoses are based on ICD-10 classification, which we have now clarified in Table 1, including reference (2).Additionally, comorbidities are collected and provided as an age-adjusted Charlson Comorbidity Index. Furthermore, outcomes are included in the registry, i.e. mortality, hospital admission, ED-revisits, hospital readmission and ICU admission. Currently, the DRAIM registry does not capture information on treatments received in the emergency department (ED). Given the complexity of acute conditions and the wide range of possible treatments, it is not feasible to collect and analyze this data comprehensively at this stage. However, as DRAIM continues to develop, we will assess whether treatment details and adherence to guidelines can be incorporated into the registry. We have clarified this in the limitations section—please refer to pages 17 lines 317-321 for the updated text: ‘Furthermore, the registry does not capture information on specific treatments received in the ED. Recording all treatments in a standard quality registry is not feasible due to the vast number and complexity of treatment options, as well as the lack of standardized documentation in EHRs. However, as DRAIM continues to develop, we will assess whether treatment details and adherence to guidelines can be incorporated into the registry’.

Comment 1.3: Is it too broad to include all diagnosis in internal medicine? The board criteria will not focus in the specific treatment and outcome.

Reply 1.3: Thank you for your comment. As mentioned in our response above, the DRAIM registry does includes all ICD-10 diagnoses within internal medicine, but it does not yet capture treatment details for these conditions. Despite that we did not report ICD-10 diagnosis, the quality registry does allow for analysis of diagnoses using either ICD-10 codes or presenting complaints.

Comment 1.4: Please begin the discussion with the main findings of your study.

Reply 1.4: We agree with your comment and have revised the discussion to begin with the main findings of our study. See pages 14-15, lines 248-253: This DRAIM-cohort study shows that approximately half of the patients presenting for internal medicine in the ED were aged ≥65 years, and one-third presented with general malaise. Patients with general malaise were more often older, female, had more comorbidities, and were triaged into lower urgency categories. Additionally, they experienced more adverse outcomes, including longer LOS-ED, higher hospital admission rates, but lower ICU admission rates.’

Comment 1.5: Please elaborate more the limitation of the study.

Reply 1.5: Thank you for your comment. We have expanded the limitations section to provide a more detailed discussion, as mentioned in our previous responses. Specifically, we have clarified the constraints of the DRAIM registry, including the lack of treatment data and the broad scope of internal medicine diagnoses. Additionally, we have outlined future plans for more focused analyses of specific diseases and their treatments. Please see the revised limitations section on pages 17, lines 308-323: ‘Limitations of our study include the following: firstly, a minority of variables in this cohort were incomplete. However, we try to minimize incomplete variables, by providing the local data collection team with feedback reports on missing data. Secondly, data extraction relies on the data registry in the electronic health records (EHR), where diagnoses are primarily based on financial records. This approach can lead to inaccuracies, as seen in cases where patients presenting with acute exacerbations of chronic conditions may be classified under their underlying chronic illness rather than the acute episode itself. Similarly, in the center used for this cohort, patients referred to the ED by their primary medical specialist were registered as self-referral, making it difficult to distinguish them from true self-referred patients. Additionally, while the registry uses ICD-10 coding, we have further grouped diagnoses into broader categories to facilitate meaningful data analysis. This aids the identification of trends, but may also contribute to a loss of diagnostic granularity. Furthermore, the registry does not capture information on specific treatments received in the ED. Recording all treatments in a standard quality registry is not feasible due to the vast number and complexity of treatment options, as well as the lack of standardized documentation in EHRs. However, as DRAIM continues to develop, we will assess whether treatment details and adherence to guidelines can be incorporated into the registry. Lastly, data is gathered and analyzed with a delay of at least one month. Providing real-time insights would enhance the value of the reported information, allowing for immediate adjustments in the organization of care.

Comment 1.6: The conclusion should be concise and clear, and only answer the research question.

Reply 1.6: We agree with your comment and have revised the conclusion, adding a subheading to enhance clarity and ensure it directly answers the research question. See pages 17-18, lines 326-329: DRAIM provides valuable insights and identifying bottlenecks in the patient journey throughout the acute care pathway, facilitating the evaluation and improvement of acute care quality. We have demonstrated that older patients presenting to the ED with poorly defined symptoms experience more adverse outcomes, particularly longer LOS-ED and higher hospital admission rates.

1. Nemec M, Koller MT, Nickel CH, Maile S, Winterhalder C, Karrer C, et al. Patients presenting to the emergency department with non-specific complaints: the Basel Non-specific Complaints (BANC) study. Acad Emerg Med. 2010;17(3):284-92.

2. World Health O, Pan American Health O. The ICD-10 classification of mental and behavioural disorders: Diagnostic criteria for research. 1993.

---

## [Decision Letter · Decision Letter 1]

9 Nov 2025

PONE-D-25-05039R1Introducing the Dutch Quality Registry for Acute Internal Medicine (DRAIM): Method of development and opportunities of use.PLOS ONE

Dear Dr.  van der Velde,

Thank you for submitting your manuscript to PLOS ONE. After careful consideration, we feel that it has merit but does not fully meet PLOS ONE’s publication criteria as it currently stands. Therefore, we invite you to submit a revised version of the manuscript that addresses the points raised during the review process.

We look forward to receiving your revised manuscript.

Kind regards,

Rehab Al-Ansari

Academic Editor

PLOS ONE

Journal Requirements:

Additional Editor Comments:

**please indorse reviewer comments carefully**

Dear Author,

it's interesting topic which needs to highlight to improve the quality of patient care.

the reviewer comments were appropriately indorsed in revision version.

however further minor revision is needed .

Best regards,

Dr. Rehab Al-Ansari

Reviewers' comments:

Reviewer's Responses to Questions

**Comments to the Author**

1. If the authors have adequately addressed your comments raised in a previous round of review and you feel that this manuscript is now acceptable for publication, you may indicate that here to bypass the “Comments to the Author” section, enter your conflict of interest statement in the “Confidential to Editor” section, and submit your "Accept" recommendation.

Reviewer #1: All comments have been addressed

Reviewer #2: All comments have been addressed

2. Is the manuscript technically sound, and do the data support the conclusions?

Reviewer #1: Partly

Reviewer #2: Yes

3. Has the statistical analysis been performed appropriately and rigorously? 

Reviewer #1: Yes

Reviewer #2: No

4. Have the authors made all data underlying the findings in their manuscript fully available?

Reviewer #1: Yes

Reviewer #2: Yes

5. Is the manuscript presented in an intelligible fashion and written in standard English?

Reviewer #1: Yes

Reviewer #2: Yes

6. Review Comments to the Author

Reviewer #1: Hello

The article is appropriate and has examined the details to an acceptable extent. In the conclusion section, a few lines should be mentioned to provide suggestions for future studies.

Good luck.

Reviewer #2: Methodology:

Weaknesses and risks

Design ambiguity:

The paper calls itself “multicentre” but only one site is analyzed. It reads as both registry description and single-centre cohort. Clarify scope: if this is a pilot, label it so throughout.

Representativeness:

One teaching hospital’s data (Veldhoven) can’t yet reflect national quality or feasibility. This limits external validity; should be made explicit.

Outcome inference:

It’s descriptive, yet they imply causal relationships (“older patients with general malaise experience more adverse outcomes”). Without multivariate adjustment, confounding (age, comorbidity, triage category) could fully explain this.

Statistical parsimony:

Solely bivariate comparisons with p-values. With 6,000+ subjects, most comparisons reach significance even for trivial effects. A short multivariable model (e.g., logistic regression for hospital admission vs. presenting complaint) would have added real analytic value.

Data quality assurance:

They mention quarterly sampling and feedback but provide no quantitative quality indicators (e.g., % missing per variable, inter-rater reliability for complaint coding).

Ethics and patient involvement:

Well described, though “no access to participants” sounds overstated. It’s automated EHR extraction, but authors still accessed de-identified records; wording could be clearer.

Data availability:

“Available upon reasonable request via Dutch website” meets PLOS’s minimal criteria but is technically restricted, not open; this may cause a query from the editor unless justified (GDPR).

Structure and language:

Clarity and readability

English is generally clear but stiff; reads like a translated policy report. Occasional Dutch-style phrasing (“restructuring the acute care chain”) could be Anglicised (“restructuring acute-care pathways”).

The paper could lose ~10 % of text without losing meaning; repetition between Abstract, Introduction, Discussion, and Conclusion.

Flow

Methods → Results → Discussion now follow journal convention, but the Methods remain very long because of Table 1. Condense the description and move full variable listing to Supplementary Appendix.

Tables

Table 2–5 are dense, with redundant p-values comparing all combinations of age groups. Simpler contrasts (trend or ANOVA) would improve readability.

Tables should include n per column and missing data footnotes; a CONSORT-like flow diagram showing total ED presentations and exclusions would help context.

References

Reasonably current and appropriate. Some URLs in Dutch grey literature (e.g., LNAZ reports) are unstable; PLOS prefers DOI or archived version.

Formatting

Minor issues: inconsistent spacing (e.g., “general malaise ” vs “general malaise”), extraneous commas, and occasionally awkward pluralization (“the registry do not capture”).

To do:

1. Clarify whether this is a single-centre pilot or a multicentre dataset; revise title and abstract accordingly.

2. Add a statement acknowledging that results from one hospital cannot yet be generalized nationally.

3. Limit inferential claims (no adjustment for confounders).

4. Consider summarizing the Methods and moving detailed variable lists to supplementary material.

5. Improve English phrasing and condense repetitive text.

6. Ensure the data-sharing statement meets PLOS ONE’s open-data policy (explain the legal restriction explicitly).

7. Simplify the Results tables for readability.

Minor editorial suggestions

- Replace “restructuring the acute care chain” with “restructuring acute-care pathways”.

- Rephrase “provides valuable insights and identifying bottlenecks” → “provides valuable insights and identifies bottlenecks”.

- Check consistency of abbreviations (LOS-ED, CCI, NSC).

7. PLOS authors have the option to publish the peer review history of their article (what does this mean?). If published, this will include your full peer review and any attached files.

Reviewer #1: No

Reviewer #2: **Yes:** Inge Roggen

---

## [Author Response · Author response to Decision Letter 2]

8 Mar 2026

Dear Editor and Reviewers,

We sincerely thank you for your thorough review and constructive feedback. We appreciate the time and effort you invested in improving the quality of our manuscript. Below, we provide a detailed response to each comment. All changes have been incorporated into the revised manuscript, and tracked changes are visible in the marked-up version.

We hope that the revisions adequately address all concerns raised by the reviewers and the editor. Thank you again for your valuable feedback and for the opportunity to improve our manuscript.

Kind regards,

Marleen van der Velde & Elsemieke Mols

Reviewer #1:

The article is appropriate and has examined the details to an acceptable extent. In the conclusion section, a few lines should be mentioned to provide suggestions for future studies.

Response:

Thank you for this helpful suggestion. We have added a paragraph in the Conclusion section outlining recommendations for future research, including multicentre validation and expanded analyses. ‘‘Future use of DRAIM for multicentre studies will facilitate the evaluation of impact of heterogeneous care pathways for older patients with nonspecific complaints in acute care.’ (See page 17-18, lines 336-8)

Reviewer #2

Comment 1:

“Clarify whether this is a single-centre pilot or a multicentre dataset; revise title and abstract accordingly.”

Response:

Thank you for highlighting this important point. We have revised both the title and the abstract to clearly indicate that this study reports findings from a single-centre pilot within the broader multicentre DRAIM initiative.

• Title change:

The title now reads:

“Introducing the Dutch Quality Registry for Acute Internal Medicine (DRAIM): Method of development and opportunities of use from a single-centre pilot study.”

• Abstract changes:

In the Methods section of the abstract, we explicitly state:

“DRAIM is a multicenter quality registry. This study reports findings from a single-centre pilot conducted in one teaching hospital.”

(See page 2, lines 38-39.)

Comment 2:

“Representativeness: One teaching hospital’s data can’t yet reflect national quality or feasibility. This limits external validity; should be made explicit.”

Response:

We agree and have added a statement the abstract and the discussion acknowledging this limitation and clarifying that the findings cannot be generalized nationally at this stage.

• Abstract changes

While findings cannot yet be generalized nationally, they highlight important trends…

(See page 3, lines 53-54)

• Discussion changes

While findings cannot yet be generalized nationally, they highlight important trends, particularly the vulnerability of older patients and those presenting with non-specific complaints.

(See page 15, lines 262-264)

Comment 3:

“Statistical parsimony: Solely bivariate comparisons with p-values. Consider adding a short multivariable model.” Limit inferential claims (no adjustment for confounders).

Response:

We thank the reviewer for this comment. In the revised manuscript, we have expanded the statistical analysis to include multivariable regression models in addition to descriptive and bivariate analyses. Specifically, we performed multivariable linear regression to assess the association between presenting complaint and emergency department length of stay, and multivariable logistic regression to analyze hospital admission. Both models were adjusted a priori for clinically relevant confounders, including age, sex, urgency category (Netherlands Triage System), comorbidity burden, mode of arrival, and number of medications used prior to the ED visit.

- Statistical method changes

The association between presenting complaint and emergency department length of stay (LOS-ED) was assessed using multivariable linear regression, with log-transformed LOS-ED due to right-skewness. Results are reported as regression coefficients (B) with 95% confidence intervals (CI). Hospital admission was analyzed using multivariable logistic regression, with results reported as odds ratios (ORs) and 95% confidence intervals. Both models were adjusted a priori for age, sex, urgency category (Manchester Triage System (MTS), U0–U5, with U0 indicating the highest and U5 the lowest urgency), comorbidity burden as measured by Charlson Comorbidity Index (CCI), mode of arrival and number of medications used prior to the ED visit. MTS and CCI were entered as categorical variables with U3 (moderate urgency) and no comorbidity (CCI=0) [MK1.1]as reference categories; mode of arrival was dichotomized as ambulance versus own transport. Linearity of age was assessed by including a quadratic age term, and effect modification by presenting complaint was evaluated using an interaction term between age and general malaise. As neither term was statistically significant, they were not retained in the final models. Multicollinearity was assessed using variance inflation factors. Model fit for the linear regression was assessed using adjusted R². For the logistic regression model, model fit was assessed using −2 log likelihood, Nagelkerke R², and the Hosmer–Lemeshow test.

See page 7-8, lines 147-161

- Description of results

The Results section has been reorganized to improve readability with the inclusion of these analyses, see page 8-14, lines 169-248. For the addition of the multivariable analysis, see page 13-14, lines 231-248.

‘Association of presenting complaints with ED outcomes

In multivariable analyses (table 5), general malaise was independently associated with both prolonged LOS-ED and hospital admission. After adjustment for age, sex, urgency category, comorbidity burden, mode of arrival and medication use, patients presenting with general malaise had a longer LOS-ED (B = 0.046, 95% CI 0.022–0.070), corresponding to an approximately 4.7% increase in LOS-ED.

In addition, general malaise was associated with a 51% higher odds of hospital admission (OR 1.51, 95% CI 1.34–1.71). Increasing age, higher urgency categories, and arrival by ambulance were also independently associated with both outcomes.

The linear regression model explained 3.4% of the variance in LOS-ED, while the logistic regression model explained 21.3% of the variance in hospital admission. No evidence of multicollinearity was observed. Although the Hosmer–Lemeshow test indicated imperfect calibration for the admission model, this is likely attributable to the large sample size.’

Accordingly, inferential statements in the manuscript have been revised to reflect adjusted associations rather than unadjusted bivariate comparisons, and causal language has been avoided.

Comment 4:

“Simplify the result section for better readability. Tables 2–5 are dense, with redundant p-values comparing all combinations of age groups. Simpler contrasts (trend or ANOVA) would improve readability. Tables should include n per column and missing data footnotes; a CONSORT-like flow diagram showing total ED presentations and exclusions would help context.”

Response:

Thank you for these valuable suggestions. We have revised table 2-3 (now table 1 and 2) by removing the pairwise p-values and presenting a single overall p-value per variable using ANOVA for continuous variables and chi-square tests for categorical variables. We have also added an n-column and included footnotes summarizing missing values. The Results section has been reorganized to improve readability with the inclusion of these analyses, see page 8-14, lines 169-248. Furthermore, we have added a flow diagram (Figure 1) to illustrate patient selection and exclusions, providing additional context for the cohort. (See page 8, lines 169-173).

Figure 1: Patient Selection and Cohort Flow

Comment 5:

“Consider summarizing the Methods and moving detailed variable lists to supplementary material.”

Response:

Thank you for this helpful suggestion. We have substantially revised the Methods section to provide a concise overview. We also moved the detailed list of variables to the Supplementary Table S1. (See pages 5-7, lines 97-168).

Comment 6:

“Improve English phrasing and condense repetitive text.”

Response:

We have thoroughly revised the manuscript for clarity and conciseness, reducing redundancy and improving phrasing as suggested. Examples include replacing “restructuring the acute care chain” with “restructuring acute-care pathways” and correcting minor grammatical inconsistencies. (See throughout manuscript.)

Comment 7:

“Ensure the data-sharing statement meets PLOS ONE’s open-data policy.”

Response:

We have updated the data-sharing statement to explicitly explain GDPR-related restrictions and confirm that data are available upon reasonable request via the Dutch registry website. (See page 19, lines 359-360.)

---

## [Decision Letter · Decision Letter 2]

10 May 2026

Introducing the Dutch Quality Registry for Acute Internal Medicine (DRAIM):

PONE-D-25-05039R2

Dear Dr. M.G.A.M. van der Velde,

We’re pleased to inform you that your manuscript has been judged scientifically suitable for publication and will be formally accepted for publication once it meets all outstanding technical requirements.

Kind regards,

Rehab Al-Ansari

Academic Editor

PLOS One

Additional Editor Comments (optional):

Reviewers' comments:

Reviewer's Responses to Questions

**Comments to the Author**

1. If the authors have adequately addressed your comments raised in a previous round of review and you feel that this manuscript is now acceptable for publication, you may indicate that here to bypass the “Comments to the Author” section, enter your conflict of interest statement in the “Confidential to Editor” section, and submit your "Accept" recommendation.

Reviewer #2: All comments have been addressed

2. Is the manuscript technically sound, and do the data support the conclusions?

Reviewer #2: Yes

3. Has the statistical analysis been performed appropriately and rigorously? 

Reviewer #2: Yes

4. Have the authors made all data underlying the findings in their manuscript fully available?

Reviewer #2: Yes

5. Is the manuscript presented in an intelligible fashion and written in standard English?

Reviewer #2: Yes

6. Review Comments to the Author

Reviewer #2: Thank you, all the comments have been adressed

7. PLOS authors have the option to publish the peer review history of their article (what does this mean?). If published, this will include your full peer review and any attached files.

Reviewer #2: **Yes:** Inge Roggen

---

## [Editor Report · Acceptance letter]

PONE-D-25-05039R2

PLOS One

Dear Dr. van der Velde,

I'm pleased to inform you that your manuscript has been deemed suitable for publication in PLOS One. Congratulations! Your manuscript is now being handed over to our production team.

Kind regards,

on behalf of

Dr. Rehab Al-Ansari

Academic Editor

PLOS One